# Artificial intelligence-enabled prediction of chemotherapy-induced cardiotoxicity from baseline electrocardiograms

Ryuichiro Yagi[1,2,3,8], Shinichi Goto [1,2,4,8], Yukihiro Himeno[5], Yoshinori Katsumata[6], Masahiro Hashimoto[7], Calum A. MacRae [1,2] & Rahul C. Deo [1,2] ✉

Anthracyclines can cause cancer therapy-related cardiac dysfunction (CTRCD) that adversely affects prognosis. Despite guideline recommendations, only half of the patients undergo surveillance echocardiograms. An AI model detecting reduced left ventricular ejection fraction from 12-lead electrocardiograms (ECG) (AI-EF model) suggests ECG features reflect left ventricular pathophysiology. We hypothesized that AI could predict CTRCD from baseline ECG, leveraging the AI-EF model's insights, and developed the AI-CTRCD model using transfer learning on the AI-EF model. In 1011 anthracycline-treated patients, 8.7% experienced CTRCD. High AI-CTRCD scores indicated elevated CTRCD risk (hazard ratio (HR), 2.66; 95% CI 1.73–4.10; log-rank $p < 0.001$). This remained consistent after adjusting for risk factors (adjusted HR, 2.57; 95% CI 1.62–4.10; $p < 0.001$). AI-CTRCD score enhanced prediction beyond known factors (time-dependent AUC for 2 years: 0.78 with AI-CTRCD score vs. 0.74 without; $p = 0.005$). In conclusion, the AI model robustly stratified CTRCD risk from baseline ECG.

Cancer therapy-related cardiac dysfunction (CTRCD) occurs in more than 10% of patients treated with cardiotoxic agents such as anthracyclines[1–4] and is strongly associated with poor prognosis[5,6]. If adequately treated before the onset of overt heart failure, reduced left ventricular ejection fraction (LVEF) due to chemotherapy is often reversible, and clinical outcomes can be improved[7]. Thus, current guidelines strongly recommend surveillance echocardiograms for all patients treated with cardiotoxic regimens[8–12]. However, in the real-world setting, performing echocardiograms on all the patients may not always be feasible for various reasons, including the lack of integrated systems for such surveillance, the limited capacity for echocardiograms at some institutions, and the lack of awareness of

the extent of the risk of CTRCD[13]. In fact, it has been reported that approximately half of the patients at risk do not receive echocardiograms during cardiotoxic cancer treatment[14–16]. Although accurate risk assessment before systemic cancer treatment could facilitate physicians' detection of the occurrence of CTRCD, its prediction remains a major challenge due to the limited predictive accuracy and availability of current approaches[8,17–19]. If a screening strategy utilizing a more accessible modality capable of accurate stratification of CTRCD risk is established to triage the patients to surveillance echocardiography under resource constraints, then fewer CTRCD patients would be missed with similar echocardiography resource utilization.

[1]One Brave Idea and Division of Cardiovascular Medicine, Department of Medicine, Brigham and Women's Hospital, Boston, MA, USA. [2]Harvard Medical School, Boston, MA, USA. [3]Department of Preventive Medicine and Public Health, Keio University School of Medicine, Tokyo, Japan. [4]Division of General Internal Medicine & Family Medicine, Department of General and Acute Medicine, Tokai University School of Medicine, Isehara, Kanagawa, Japan. [5]Department of Cardiology, Keio University School of Medicine, Tokyo, Japan. [6]Institute for Integrated Sports Medicine, Keio University School of Medicine, Tokyo, Japan. [7]Department of Radiology, Keio University School of Medicine, Tokyo, Japan. [8]These authors contributed equally: Ryuichiro Yagi, Shinichi Goto. ✉e-mail: rdeo@bwh.harvard.edu

12-lead electrocardiogram (ECG) is a non-invasive, inexpensive, and accessible modality for cardiac evaluation. Recently, artificial intelligence (AI) algorithms analyzing a single 12-lead ECG demonstrated considerable potential to improve diagnostic accuracy for cardiac abnormalities beyond human physician recognition[20–23]. In these settings, an AI model accurately detected reduced LVEF from ECGs (AI-EF model)[24], suggesting that AI could interpret subtle ECG abnormalities that are deeply associated with cardiac systolic dysfunction. We hypothesized that the features might be shared in those who are susceptible to cardiotoxicity but do not yet have reduced LVEF, and thus the AI-EF model could be repurposed to evaluate the risk of CTRCD in deployment to assess potential or subclinical cardiac systolic dysfunction (undetected on routine echocardiography) from ECG via transfer learning against the occurrence of CTRCD. To test this hypothesis, we assessed whether the AI algorithm, trained by taking the AI-EF model as the pre-trained model, is capable of stratifying the risk for CTRCD from baseline ECG in patients treated with anthracyclines.

## Results

### Patient population

A total of 5495 patients received chemotherapy with a regimen including anthracyclines at Brigham and Women's Hospital (BWH) and Massachusetts General Hospital (MGH) between June 1st, 2015 and October 1st, 2020. Of these, 1138 individuals who underwent ECG and transthoracic echocardiogram (TTE) within 90 days prior to the initial treatment with chemotherapy were retrospectively identified (Supplementary Fig. 1, Supplementary Table 1, 2). Of note, even at these heavily resourced academic medical centers, nearly half of the at-risk population did not have a baseline echocardiogram documented, though many may have had such testing at external referral institutions. The median follow-up duration was 560 days (Interquartile range, 149 to 999). In total, 99 participants experienced CTRCD, defined as LVEF < 53% and >10% decrease in LVEF from the baseline[25].

Similarly, a total of 880 cancer patients were treated with anthracyclines at Keio University Hospital between January 2013 and December 2019. Of those, 190 patients who underwent baseline ECG and TTE were included in the study (Supplementary Table 3). To construct a training dataset, the BWH cohort was randomly split in a 2:8 ratio, and the former ($n = 127$) was used with the entire dataset from the Keio cohort as the training dataset (Supplementary Fig. 1, Supplementary Table 4).

Overall, participants in the test set were $57.1 \pm 16.4$ years old, and 47.8% were male (Table 1). Most cancer diagnoses were hematologic malignancies ($n = 704$, 69.7%). The mean LVEF at baseline was $65.1 \pm 6.5$%. While mean age was not different between patients with and without CTRCD, baseline LVEF was significantly lower at baseline in patients who went on to develop CTRCD than in patients without CTRCD (age, $57.5 \pm 16.8$ years and $57.1 \pm 16.3$ years, $p = 0.97$; LVEF, $62.4 \pm 6.5$% and $65.4 \pm 6.4$%, $p < 0.001$ for patients with and without CTRCD, respectively). The prevalence of comorbidities was similar between the two groups. Patients at BWH were older than those at MGH, and a lower baseline LVEF and higher prevalence of leukemia were observed at BWH compared with MGH.

### AI-CTRCD model predicts the risk of CTRCD from ECG taken prior to chemotherapy initiation

If ECG-based baseline screening accurately stratifies the risk of CTRCD, fewer patients with CTRCD will be missed when the same number of echocardiograms are performed. We, therefore, sought to predict CTRCD from ECG prior to the initiation of chemotherapy. However, collecting sufficient data to train a *de-novo* deep-learning model was impractical given the limited number of those who were treated with regimens including anthracyclines and developed CTRCD. Thus, we pursued an alternative approach driven by the hypothesis that our

previously published general AI-EF model could be repurposed to evaluate the risk of CTRCD using transfer learning[26]. In brief, our AI-EF model achieved excellent discrimination for decreased cardiac systolic function (LVEF less than 40%) with an area under the receiver operating curve (AUROC) of 0.91 in the internal test datasets and was well-generalized on datasets from three external institutions. We thus updated this AI-EF model by additionally training on ECGs obtained from patients treated with anthracyclines and tested the ability of the model to stratify the risk of CTRCD occurring during subsequent follow-up.

In line with our hypothesis, our AI-CTRCD model stratified the risk of CTRCD using baseline ECG. Patients in the high AI-CTRCD score group were at higher risk for developing CTRCD compared to those in

**Table 1 | Baseline demographics and clinical characteristics of the MGB test cohort, stratified by the occurrence of CTRCD**

| | | CTRCD | | |
|---|---|---|---|---|
| | All (*n* = 1011) | No (*n* = 923) | Yes (*n* = 88) | *P*-value |
| Age, years (SD) | 57.1 (16.4) | 57.0 (16.3) | 57.1 (16.8) | 0.966 |
| Male, *n* (%) | 483 (47.8) | 434 (47.0) | 49 (55.7) | 0.149 |
| Race, *n* (%) | | | | 0.053 |
| White | 849 (84.0) | 784 (84.9) | 65 (73.9) | |
| Black | 39 (3.9) | 33 (3.6) | 6 (6.8) | |
| Asian | 38 (3.8) | 32 (3.5) | 6 (6.8) | |
| Other | 85 (8.4) | 74 (8.0) | 11 (12.5) | |
| Diagnosis, *n* (%) | | | | 0.004 |
| Lymphoma | 395 (39.1) | 363 (39.3) | 32 (36.4) | |
| Leukemia | 309 (30.6) | 268 (29.0) | 41 (46.6) | |
| Breast cancer | 141 (13.9) | 137 (14.8) | 4 (4.5) | |
| Sarcoma | 89 (8.8) | 82 (8.9) | 7 (8.0) | |
| Other | 77 (7.6) | 73 (7.9) | 4 (4.5) | |
| LVEF, % (SD) | 65.1 (6.5) | 65.4 (6.4) | 62.4 (6.5) | <0.001 |
| Comorbidities, *n* (%) | | | | |
| CAD | 52 (5.1) | 43 (4.7) | 9 (10.2) | 0.045 |
| Hypertension | 294 (29.1) | 266 (28.8) | 28 (31.8) | 0.639 |
| Diabetes | 103 (10.2) | 93 (10.1) | 10 (11.4) | 0.844 |
| Dyslipidemia | 352 (34.8) | 316 (34.2) | 36 (40.9) | 0.255 |
| Obesity | 333 (32.9) | 302 (32.7) | 31 (35.2) | 0.719 |
| Smoke, *n* (%) | 475 (47.0) | 434 (47.0) | 41 (46.6) | 1 |
| Initial anthracycline dose, mg/m², median [IQR] | 40.0 [25.0, 50.0] | 40.0 [25.0, 50.0] | 45.0 [30.0,50.0] | 0.36 |
| Cumulative anthracycline dose, mg/m², median [IQR] | 180 [90, 285] | 180 [90, 281] | 186 [95, 291] | 0.17 |
| ECG abnormalities, *n* (%) | | | | |
| 1AVb | 20 (2.0) | 17 (1.8) | 3 (3.4) | 0.543 |
| Atrial fibrillation | 25 (2.5) | 21 (2.3) | 4 (4.5) | 0.342 |
| RBBB | 19 (1.9) | 17 (1.8) | 2 (2.3) | 1 |
| LBBB | 13 (1.3) | 9 (1.0) | 4 (4.5) | 0.019 |
| ST-T change | 6 (0.6) | 6 (0.7) | 0 (0.0) | 0.974 |
| LVH | 85 (8.4) | 81 (8.8) | 4 (4.5) | 0.244 |
| PAC | 58 (5.7) | 52 (5.6) | 6 (6.8) | 0.828 |
| PVC | 16 (1.6) | 13 (1.4) | 3 (3.4) | 0.322 |
| Low voltage | 9 (0.9) | 9 (1.0) | 0 (0.0) | 0.736 |
| Paced rhythm | 11 (1.1) | 9 (1.0) | 2 (2.3) | 0.56 |
| Any abnormality | 232 (22.9) | 207 (22.4) | 25 (28.4) | 0.253 |

A χ2 test (two-sided) was used to identify differences in proportions across race and diagnosis. A Wilcoxon test (two-sided) was used for initial anthracycline dose and cumulative anthracycline dose, and a *t* test (two-sided) was used for other variables. No adjustments were made for multiple comparisons.
*CTRCD* cancer therapy-related cardiac dysfunction, *LVEF* left ventricular ejection fraction, *CAD* coronary artery disease, *ECG* electrocardiogram, *1AVb* first degree atrio-ventricular block, *RBBB* right bundle branch block, *LBBB* left bundle branch block, *LVH* left ventricular hypertrophy, *PAC* premature atrial contraction, *PVC* premature ventricular contraction, *SD* standard deviation, *IQR* interquartile range. N represents number of patients.

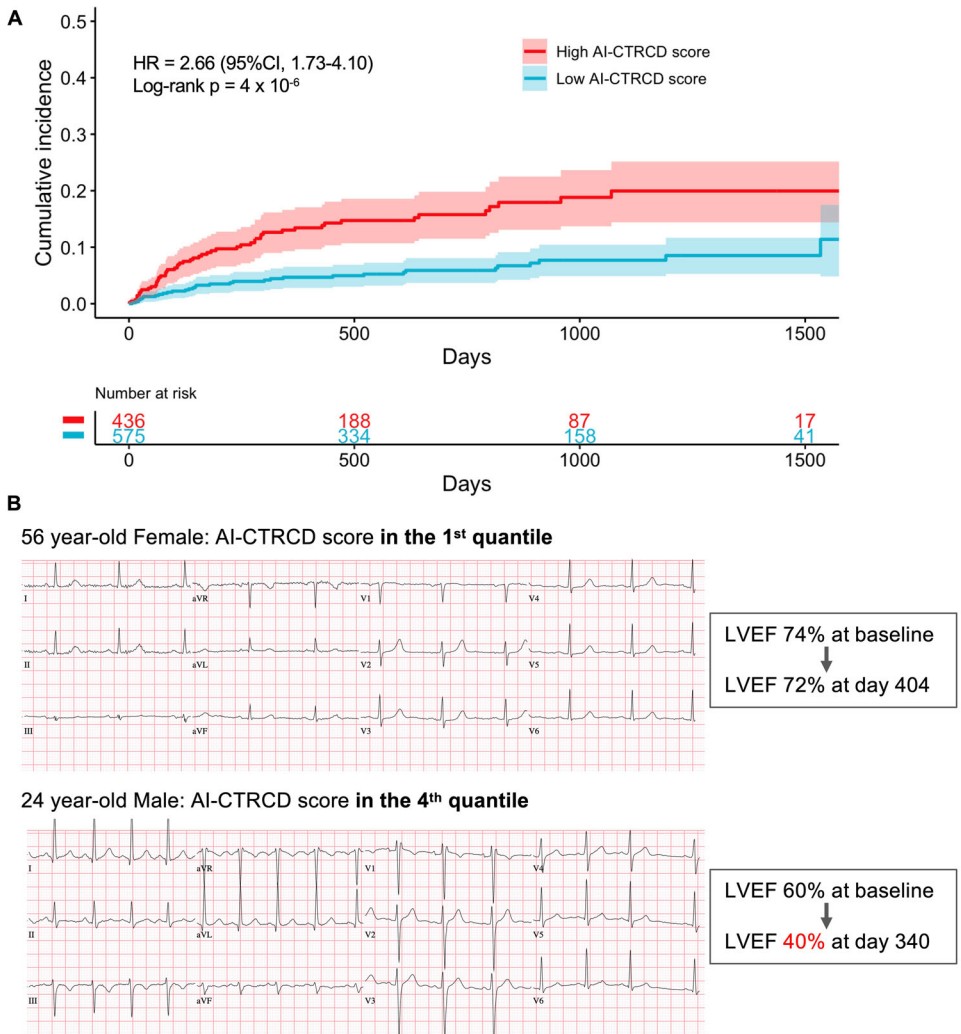

**Fig. 1 | Comparison of high and low AI-CTRCD score groups. A** Kaplan–Meyer plot showing cumulative incidence of CTRCD between high and low AI-CTRCD score. The transparent ribbons indicate 95%CI. Source data are provided as a Source Data file. **B** Representative electrocardiograms of patients with high and low AI-CTRCD score. AI artificial intelligence, CTRCD cancer therapy-related cardiac dysfunction, HR hazard ratio, CI confidence interval, LVEF left ventricular ejection fraction.

the low AI-CTRCD score group (incidence rate per 100 person-year, 8.93 vs 3.08 in high and low AI-CTRCD score groups, respectively; hazard ratio (HR), 2.66; 95% confidence interval (CI), 1.73–4.10; log-rank $p < 0.001$; Fig. 1A). This finding was consistent after adjustment for known risk factors including age, sex, race, cancer types, low baseline observed LVEF on echocardiogram, comorbidities such as coronary artery disease and hypertension[2,26–28], and the presence of overt ECG abnormalities (adjusted HR, 2.57; 95%CI, 1.62–4.10; $p < 0.001$; Fig. 2).

### The AI-CTRCD model stratifies the risk of CTRCD across clinical subgroups

Subgroup analyses regarding cancer types, sex, baseline LVEF, and anthracycline dose were performed to understand the robustness of the model performance across different characteristics. First, we performed a subgroup analysis by cancer type, given their large impact on patients' general medical condition, care, and prognosis. The risk of CTRCD was consistently higher in the high AI-CTRCD score group compared to those in the low AI-CTRCD score group regardless of the cancer type (HR 2.52; 95%CI 1.57–4.06; log-rank $p < 0.001$ and HR 2.91; 95%CI 1.03–8.18, log-rank $p = 0.03$ in patients with hematologic malignancies and with solid tumors, respectively; Fig. 3A, B).

Sex difference in cardiovascular outcomes is well recognized. We thus performed a subgroup analysis based on patients' sex, showing consistent performance of the AI-CTRCD model (HR 2.54; 95%CI 1.41–4.58; log-rank $p = 0.001$ and HR 2.70; 95%CI 1.43–5.12; log-rank $p = 0.002$, respectively; Fig. 3C, D).

Since baseline LVEF by echocardiogram was different between patients who subsequently developed CTRCD and those who did not, subgroup analysis was performed for patients with preserved and reduced LVEF at baseline. This analysis consistently demonstrated that individuals with high AI-CTRCD scores were at increased risk of CTRCD regardless of the existence of reduced LVEF at baseline (HR 2.26; 95%CI 1.30–4.00; log-rank $p = 0.003$ and HR 2.93; 95%CI 1.45–5.92; log-rank $p = 0.002$ in patients with baseline LVEF of >60% and ≤60%, respectively; Fig. 3E, F), supporting the assertion that prediction of CTRCD was not driven by our model simply detecting low-EF at the initial evaluation. This finding was robust to the LVEF values used to define reduced LVEF at baseline (Supplementary Fig. 2A–C). Furthermore, sensitivity analyses revealed that the model robustly predicted CTRCD defined using different LVEF cutoff points (Supplementary Fig 2D–F).

A high cumulative anthracycline dose is associated with the occurrence of CTRCD; however, the cumulative dose cannot be determined at the initiation of the chemotherapy and is thus not

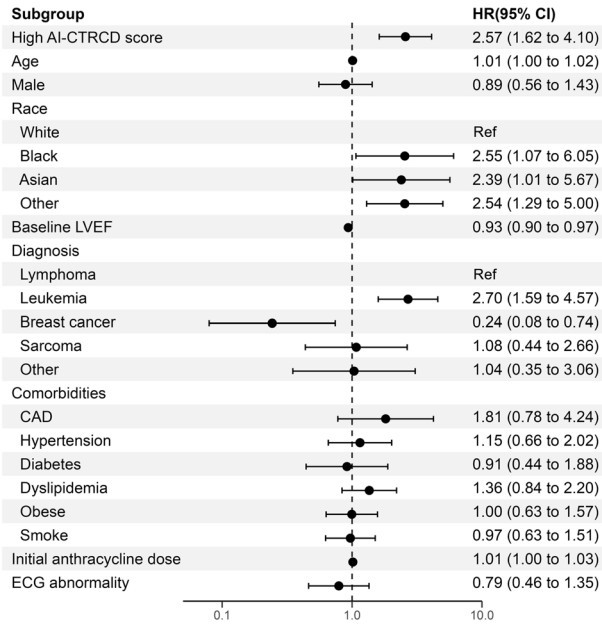

**Fig. 2 | Multivariable Cox proportional hazard analysis for developing CTRCD.** A forest plot of HRs of AI-CTRCD score and each clinical variable (n = 1101). The X-axis shows the log HRs, with error bars representing 95%CI. High AI-CTRCD score was significantly associated with CTRCD after adjusting demographic and clinical characteristics. Age, LVEF, and anthracycline dose were treated as continuous variables. Source data are provided as a Source Data file. CTRCD cancer therapy-related cardiac dysfunction, AI artificial intelligence, LVEF left ventricular ejection fraction, CAD coronary artery disease, ECG electrocardiogram, HR hazard ratio, CI confidence interval.

relevant for the prediction of adverse outcomes. Therefore, we performed stratification based on the initial anthracycline dose adjusted for body surface area as a surrogate. Using the median initial anthracycline dose across all patents (40.0 mg/m$^2$) as a stratification cutoff, the subgroup analysis revealed a similar performance of the model across the dosing strata (HR 3.48; 95%CI 1.78–6.83; log-rank p = 0.001 and HR 2.15; 95%CI 1.21–3.82; log-rank p = 0.008 in patients receiving >40 mg/m$^2$ and ≤40 mg/m$^2$ of doxorubicin equivalent anthracycline dose, respectively; Fig. 3G, H). Results were also similar when the population was stratified by cumulative anthracycline dose (HR 2.83; 95%CI 1.54–5.17; log-rank p < 0.001 and HR 2.41; 95%CI 1.30–4.49; log-rank p = 0.003 in patients receiving >180 mg/m$^2$ and ≤180 mg/m$^2$ of cumulative anthracycline dose, respectively; Supplementary Fig. 3A, B).

Additional sensitivity analyses showed the robustness of our model to a range of potential biasing factors. Model performance was consistent across two different institutions when analyzed separately (HR 2.94; 95%CI 1.58–5.00; log-rank p < 0.001 and HR 2.42; 95%CI 1.33–4.43; log-rank p = 0.003 for BWH and MGH respectively, Supplementary Fig. 3C, D). Similarly, the model robustly predicted CTRCD after excluding CTRCD within 30 days of chemotherapy (HR 2.73; 95% CI 1.70–4.41; log-rank p < 0.001; Supplementary Fig. 3E). While 42.5% of the patients did not undergo a follow-up echocardiogram, a similar result was observed even after limiting the cohort to those with at least one follow-up echocardiogram (HR 2.43; 95%CI 1.58–3.73, log-rank p < 0.001) (Supplementary Fig. 3F). Also, we found that the model prediction was not influenced by the duration between baseline ECG and TTE (Supplementary Fig. 3G, H). The results were also robust when considering death as a major competing risk using the model of Fine and Gray (sub-distribution HR, 2.37; 95%CI, 1.55–3.65; p < 0.001). Of note, patients with high AI-CTRCD scores experienced higher mortality compared to those with low AI-CTRCD scores (43.1% and 32.5% in

the high and low AI-CTRCD score group, respectively; log-rank p < 0.001; Supplementary Fig. 4).

## AI-CTRCD score significantly improved the prediction of CTRCD beyond known risk factors

To test whether the AI-CTRCD model provides additional predictive value over known risk factors, we compared prediction models using demographic and clinical variables with or without the AI-CTRCD score. Since most CTRCD cases (87.5%) were observed in a period of 2 years (consistent with previous reports[6]), time-dependent AUROCs for 0.5, 1, 1.5, and 2 years were calculated. The model, including the AI-CTRCD score consistently showed statistically higher AUROCs compared with the model without AI-CTRCD score (Table 2). The model with AI-CTRCD score detected CTRCD at 2 years with a positive predictive value (PPV) of 16.1% for a sensitivity of 91.1%, and with a PPV of 26.1% for a sensitivity of 60.5%, showing a statistically significant improvement from baseline prevalence (8.7%) (Table 3). Based on the full model containing all the clinical variables and AI-CTRCD score, patients with high scores were approximately 7 times at higher risk of CTRCD compared to those with low scores (incidence rate per 100 person-year, 9.58 vs 1.25 in the high and low score groups, respectively; Supplementary Fig. 5). The results were similar when the clinical variables were limited to those that are readily available (age, sex, race, and cancer type). The specificity and PPV were higher compared to the model without AI-CTRCD score at similar sensitivity levels, showing the independent contribution of the AI-CTRCD model for CTRCD prediction.

## The model, including the AI-CTRCD score, improves the detection of CTRCD under limited echocardiogram capacity

To test the clinical utility of the model, we performed a deployment simulation assuming a cohort of 1000 patients receiving cardiotoxic chemotherapy with the same prevalence of CTRCD (9%) in a setting where all patients undergo baseline ECG with the use of the full model (including the known risk factors and the AI-CTRCD model) to select patients on whom to perform follow-up echocardiograms. Based on the observed rate in our analysis, the resources for surveillance echocardiograms were assumed to allow 579 (57.9%) procedures in the whole simulated cohort. In this context, 52 out of 90 patients with CTRCD would be detected if the follow-up echocardiograms were randomly provided (Supplementary Table 5). In contrast, if the target population was pre-selected based on our prediction model with the cutoff for a sensitivity of 93.5% and a PPV of 14.4%, 545 patients would be allocated to undergo surveillance echocardiograms (allowing for the echocardiographic evaluation of all patients tested positive from the ECG screening tool), and 84 out of 90 CTRCD patients would be detected. This simulation suggests that a predictive tool using the baseline ECG would result in a 60% higher detection rate (an additional 32 cases) in the detection of CTRCD with the same total number of echocardiograms performed. Similarly, when using a model with AI-CTRCD score and readily available clinical variables (age, sex, race, and cancer type), 546 patients would be categorized in the high-risk group, and 81 CTRCD cases (additional 29 cases compared to the random-echo strategy) would be detected at the cutoff of a sensitivity 90.5% with a PPV 14.8%.

## Discussion
We demonstrate here that the AI-CTRCD model, trained to detect CTRCD by applying transfer learning to the AI model detecting low LVEF from ECG, accurately predicts CTRCD from baseline ECG in patients treated with cardiotoxic chemotherapy. The AI-CTRCD model robustly determined the risk of future CTRCD beyond overt ECG abnormalities and across various subgroups. We further demonstrate that the addition of the AI-CTRCD score on conventional risk factors yielded a statistically significant increment in predictive performance

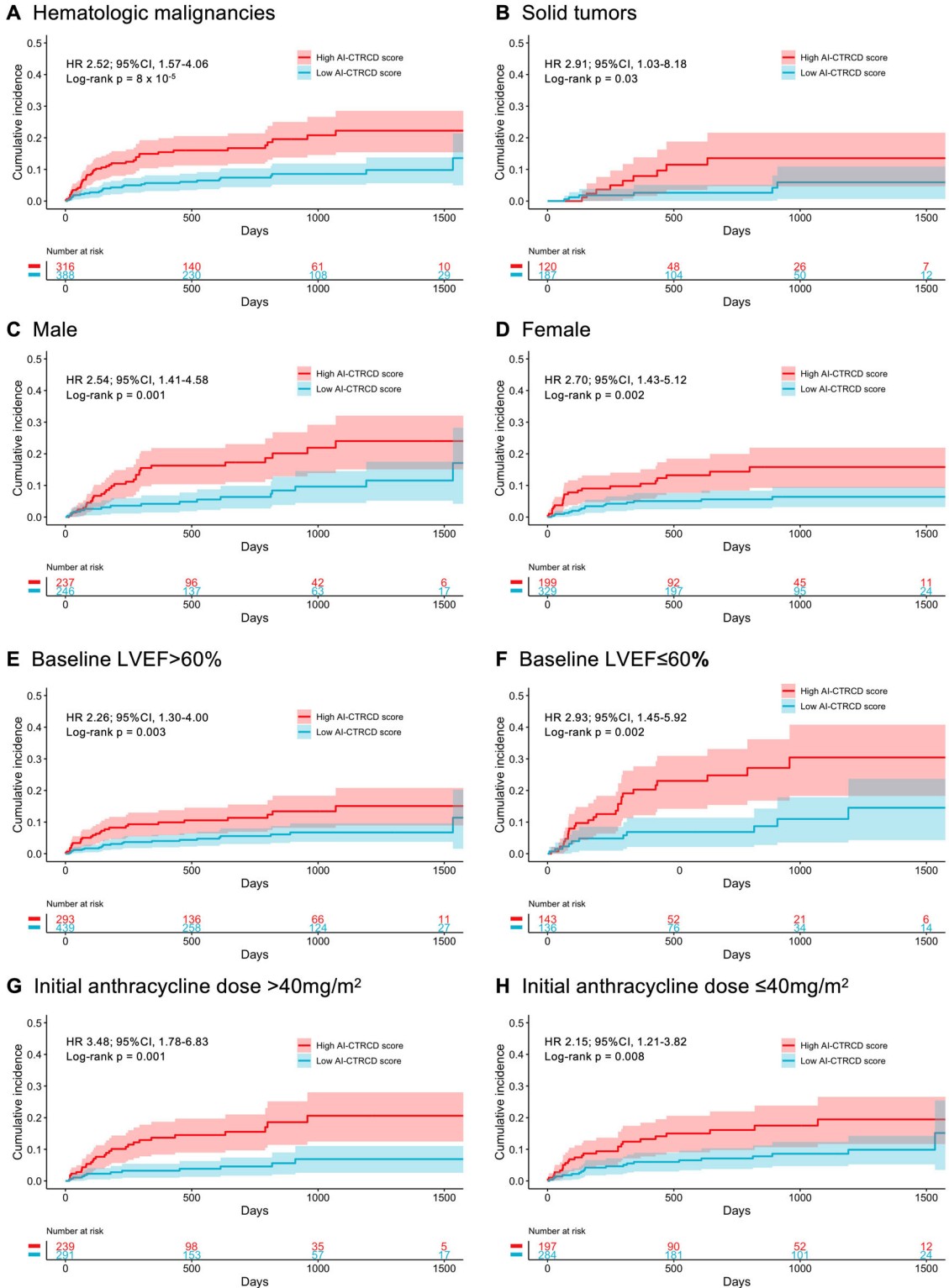

**Fig. 3 | Subgroup analyses for the risk-stratification for CTRCD using AI-CTRCD scores.** Kaplan–Meyer plots showing the cumulative incidence of CTRCD between high and low AI-CTRCD scores in those with hematologic malignancies (**A**), solid tumors (**B**), male sex (**C**), female sex (**D**), baseline LVEF > 60% (**E**), baseline LVEF ≤ 60% (**F**), and those treated with initial anthracycline dose of >40 mg/m² (**G**) and ≤40 mg/m² (**H**). The same cutoff as the primary analysis was used across all subgroups to define high and low AI-CTRCD score. Patients with high AI-CTRCD score were consistently at higher risk of CTRCD compared to patients with low AI-CTRCD score across subgroups. The transparent ribbons indicate 95%CI. Source data are provided as a Source Data file. CTRCD cancer therapy-related cardiac dysfunction, HR hazard ratio, AI artificial intelligence, LVEF left ventricular ejection fraction.

over conventional risk factors. Together these data suggest that our AI-CTRCD model captures biological information associated with vulnerability to subsequent cardiac effects from anthracyclines.

Currently, a considerable number of cancer patients receiving cardiotoxic therapeutics do not undergo follow-up echocardiograms despite guideline recommendations. A recent population-based study of 4325 patients with breast cancer who received cardiotoxic chemotherapies revealed that 46.2% of the population missed a one-year follow-up echocardiogram after initiation of treatment[14]. Consequently, a substantial number of patients with asymptomatic CTRCD may be under-detected despite the fact that even asymptomatic CTRCD is associated with poor prognosis[5,6]. Cardioprotective medical therapies have been shown to be effective in preventing the worsening or even reversing LVEF[7], highlighting the importance of optimizing screening and detection protocols to prevent the progression of CTRCD to overt heart failure. Ideally, adherence to guideline recommendations could be improved by increasing the availability of echocardiography for those receiving cardiotoxic chemotherapy. However, this approach is not always feasible given the resource constraints at most institutions[29]. An alternative approach for improving the detection while optimizing the use of current echocardiography capacity is to increase the yield of testing by risk stratification. To this end, several cardiac biomarkers have been evaluated for CTRCD prediction. For example, elevated levels of troponin or B-type natriuretic peptide from baseline after initiation of chemotherapy may identify patients at high risk for CTRCD[30]. However, it remains unclear whether baseline levels alone contribute to accurate risk assessment for CTRCD, as a consequence of the absence of definitive studies of the utility of these biomarkers. Reduced global longitudinal strain (GLS), a myocardial deformation analysis reflecting myocardial function, has been proposed as a strong predictor for CTRCD. A study of 73 patients treated with anthracycline chemotherapy reported that the AUROC for the association of baseline GLS and CTRCD was 0.77[31], while a second study found that a low baseline GLS was significantly associated with cardiac events even in those patients with normal baseline LVEF[32].

However, the availability of GLS measurement is limited due to the requirement for proprietary specialized software. Several risk scores using clinical features and treatment factors have also been developed for predicting CTRCD, none of which has been established for clinical use due to limited performance or the lack of validation studies[18,33]. Therefore, establishing a more reliable, generalizable, and accessible risk stratification strategy for CTRCD is necessary to improve the management of patients who receive cardiotoxic chemotherapy[34].

The development of an accurate CNN-based algorithm hinges on the size of the training dataset, with a substantial number of manually labeled ECG data often required[35]. However, gathering a sufficient amount of data on patients treated with anthracyclines could be impractical due to the limited number of patients along with the need for long follow-up duration. To address this challenge, we adopted a transfer learning approach. Transfer learning involves utilizing a pre-trained model for a different yet relevant problem[36] and has exhibited promising results in AI models for ECG data analysis[37]. In this study, we leveraged a CNN-based model detecting low LVEF. Our previous study showed its accurate and generalizable detection ability of low LVEF from ECG. CTRCD risk has been linked to baseline cardiac systolic function, represented by LVEF and GLS[38]. Consequently, we anticipate that the model would be capable of identifying ECG abnormalities present in patients with impaired cardiac systolic function, making it an appropriate candidate for the pre-trained model in the development of a new model to assess the baseline risk of CTRCD.

Baseline ECG, enhanced by the AI-CTRCD model, could have significant clinical utility in managing patients exposed to cardiotoxic regimens. Considering its impact on the poor outcome, it is crucial not to miss the cases with CTRCD. We proposed an AI-based approach using a single recording of baseline ECG along with easily obtainable clinical variables, as an instant tool to efficiently detect patients at high risk for CTRCD. This approach operates within the current limitation of echocardiogram capacity through prioritization of patients identified as high risk for subsequent echocardiographic monitoring. Notably, ECG is a more widely available metric demanding less expertize (particularly for data acquisition) and considerably lower cost compared to an echocardiogram. Changes in ECG parameters such as QRS prolongation can be correlated to incident CTRCD[39,40], but the association of baseline ECG measurements and risk of CTRCD has not previously been assessed. Our data demonstrate that ECG data analyzed by the AI-CTRCD model was an independent predictor of CTRCD beyond overt baseline ECG abnormalities or baseline echocardiographic data and could contribute to improved risk assessment for CTRCD. The addition of baseline ECG together with the AI-CTRCD model for general risk screening would enable accurate stratification that enables better utilization of echocardiographic resources to detect CTRCD.

Since the AI-CTRCD model is applied before the initiation of chemotherapy, the model could only detect abnormalities present at baseline. Thus, the prediction of CTRCD probably is driven by the detection of susceptibility to cardiac damage induced by

**Table 2 | Comparison of the models with and without the AI-CTRCD score for predicting CTRCD**

|  | AUROC, % (95%CI) | | ΔAUROC, % (95%CI) | P value |
|---|---|---|---|---|
|  | Model 1 | Model 2 | | |
| 0.5 year | 78.0 (72.4–83.6) | 74.2 (68.3–80.2) | 3.8 (0.4–7.1) | 0.03 |
| 1 year | 79.3 (73.8–84.8) | 75.9 (70.0–81.2) | 3.4 (0.5–6.3) | 0.02 |
| 1.5 year | 78.1 (72.6–83.7) | 74.6 (68.5–80.6) | 3.6 (0.6–6.5) | 0.02 |
| 2 years | 78.1 (72.2–84.0) | 73.8 (67.6–80.1) | 4.3 (1.3–7.2) | 0.005 |

Delong test (two-sided) was used to calculate P-values for comparing AUROCs. No adjustments were made for multiple comparisons.
Model 1: Cox model with AI-CTRCD score, Model 2: Cox model without AI-CTRCD score AUROC: area under the receiver operating curve, CI: confidence interval.

**Table 3 | Performances of the prediction model with and without the AI-CTRCD score for 2 years CTRCD at various cutoffs**

| Target sensitivity | Sensitivity | | Specificity | | PPV | | NPV | |
|---|---|---|---|---|---|---|---|---|
|  | Model 1 | Model 2 | Model 1 | Model 2 | Model 1 | Model 2 | Model 1 | Model 2 |
| 100 | | | 0 | | 8.7 | | NA | |
| 90 | 91.1 | 92.3 | 47.8 | 30.3 | 16.1 | 12.7 | 98.0 | 97.3 |
| 80 | 80.6 | 82.4 | 60.4 | 50.1 | 18.2 | 15.3 | 96.6 | 96.3 |
| 60 | 60.5 | 60.6 | 81.2 | 70.7 | 26.1 | 18.5 | 94.9 | 94.3 |

These values were calculated based on time-dependent area under the receiver operating curve for 2 years from initiation of chemotherapy. PPV at sensitivity of 100% is considered as prevalence in this cohort.
Model 1: Cox model with AI-CTRCD score, Model 2: Cox model without AI-CTRCD score.
*NA* not applicable, *PPV* positive predictive value, *NPV* negative predictive value.

chemotherapy. While the detected features may not be specific to the susceptibility to chemotherapy and could be a general marker of poor left ventricular substrate, we believe that the extraction of predictive features from a low-cost modality (i.e., ECG) is valuable. For instance, although GLS is an echocardiographic value assessing left ventricular systolic function and not necessarily specific to the risk of CTRCD, its clinical utility as a risk factor of CTRCD is widely recognized in cardio-oncology[12]. Our results indicate the feasibility of expanding the applicability of predictive measures beyond GLS by leveraging data from ECG, a much more accessible modality.

There are several limitations to this study. First, a number of patients were excluded because of a lack of ECG and/or echocardiogram prior to chemotherapy. Furthermore, CTRCD events could be missed with a lack of surveillance echocardiogram after chemotherapy. The study population might have been at a higher risk of CTRCD compared to the overall population receiving anthracyclines (physicians might have considered that these patients were likely to be at high risk for CTRCD and therefore performed ECG/echocardiographic evaluation). Nonetheless, this study indicated that the prediction model using the AI-CTRCD scores performed well in such a high-risk population. Second, the study population experienced a considerable number of deaths, which could have introduced a bias given that death is a major competing risk for CTRCD. However, the patients with high AI-CTRCD scores experienced more deaths compared to those with low AI-CTRCD scores, indicating that even if death does introduce bias, the direction of the bias would be toward the null. Furthermore, additional analysis using the Fine and Gray model demonstrated consistent results, supporting that our analyses are robust to death as a competing risk. Third, since the GLS values were not available in our dataset, it remains unclear whether the AI-CTRCD model performs better compared to the prognostic value of GLS. However, the absence of GLS measurement in the real-world setting could indicate the clinical utility of the AI-CTRCD model as an instant tool for baseline risk stratification of cardiotoxicity when access to the echocardiographic resources is limited. Fourth, another but more straightforward approach for early detection of CTRCD using the AI-EF model is to utilize surveillance ECGs taken after the initiation of chemotherapy to detect patients who already develop CTRCD as early as possible. Since routine ECG after chemotherapy is currently not recommended, patients who underwent ECGs after chemotherapy could likely have cardiac dysfunction because physicians considered them at high risk for heart failure (i.e., they presented subjective symptoms), leading to a significant bias when analyzed retrospectively. Further prospective studies are warranted to validate this strategy to enhance the potential of ECG and the AI model. Fifth, because only 20 cardiovascular disease (CVD)-related deaths in 375 patients who died during follow-up were seen in the MGB test set, the association of AI-CTRCD score and CVD-related death could not be assessed. Further studies with a larger population and longer follow-up periods are also warranted to show that the AI model could contribute to the prediction of more severe cardiac dysfunction causing CVD-related deaths.

In conclusion, we demonstrate that an AI model can be trained to detect baseline cardiac features predictive of the risk of CTRCD from baseline ECG acquired prior to cardiotoxic chemotherapy by utilizing transfer learning. The model performed well across clinically distinctive subgroups. Our findings support the clinical utility of baseline ECG, together with the AI-CTRCD model, in identifying patients treated with anthracyclines at elevated risk of CTRCD.

## Methods
### Patient selection and data collection
Patient demographics data and prescription information were retrospectively obtained from the Massachusetts General Brigham Enterprise Data Warehouse, which includes electronic health records from >20 provider locations across a large integrated delivery network in Massachusetts. Patients older than 18 years old who were treated with chemotherapy including at least one dose of anthracyclines (doxorubicin, idarubicin, daunorubicin, and epirubicin) were first identified. Among these subjects, those who had both an ECG and an echocardiogram ≤90 days before the initiation of chemotherapy were included in the study.

Patient demographics including age, sex, race, and clinical characteristic including body weight, LVEF at baseline, comorbidities (coronary artery disease, hypertension, diabetes, dyslipidemia), and prescription data were obtained from the database. The cancer diagnoses, treatment drugs, and the dose of drugs were manually confirmed by chart review. Missing variables in the dataset were also manually filled from the chart. Obesity was defined as a body mass index ≥30. Anthracycline doses were adjusted based on the following doxorubicin CTRCD equivalence: idarubicin, 5.0; daunorubicin, 0.5; epirubicin, 0.67[41].

All ECGs were recorded digitally and stored in the MUSE Cardiology Information System (GE Healthcare, U.S.), and were interpreted by physicians at each institution and then manually confirmed by a cardiologist (R.Y). ECGs were analyzed by the AI-CTRCD model as digital standard 12-lead vectorized signals.

Similarly, patient demographics data and prescription information were collected from Keio University Hospital through manual chart review with the same inclusion/exclusion criteria. ECG data were recorded and stored in a system provided by Nihon Kohden.

### AI-EF model architecture, training, and output
The development of the AI-EF model is outlined elsewhere[42]. Briefly, the model was constructed as a 2D-CNN-based model to identify LVEF ≤ 40% from 12-lead ECG voltage data as inputs. It consisted of a layer of 2D-CNN followed by 20 layers of the multi-2D-CNN module, which was constructed of 3 parallel multilayer CNNs. The ECG model was trained using data from BWH and was externally validated at three different international institutes. The evaluation of the model showed excellent discrimination of LVEF ≤ 40% (AUROC 0.91 in the BWH test set and >0.90 in all three independent institutions). The model generates a score of zero to one from 12-lead ECG voltage data. A high score (close to one) indicates a higher probability of the patient having low LVEF, whereas a low score (close to zero) indicates a prediction for normal LVEF. Since a minor number of ECGs in the study population were included in the dataset for developing the AI-EF model, we developed a new AI-EF model after excluding these ECGs from the training dataset and found the new AI-EF model had equivalent accuracy in detecting low LVEF from (Supplementary Table 6).

### Development of AI-CTRCD model
To update the model specifically for the CTRCD prediction purpose, we employed a transfer learning approach[37]. The BWH cohort was randomly split in a 2:8 ratio and the former was combined with the Keio cohort to construct a training set. The training set was further randomly split into two groups (derivation and validation sets) in a 7:3 ratio. The weights of the abovementioned AI-EF model were used as initial weights of a new AI model with the same architecture, and the model was re-trained on the new training dataset to predict the future occurrence of CTRCD (AI-CTRCD model) with a learning rate of 0.00001 using RMSprop optimizer. The training process was completed in 150 epochs, and a final mode with the highest AUROC in the validation set was chosen and tested once using the rest of the MGB set as the test cohort.

### Evaluation of the AI-CTRCD model to stratify the risk of CTRCD
The AI-CTRCD model was applied to the 12-lead ECG obtained immediately prior to the initiation of chemotherapy to calculate the AI-CTRCD scores. The study population was classified into two groups based on AI-CTRCD score using a cutoff value determined by Youden

index[43]. The main endpoint of the study was the occurrence of CTRCD during follow-up. We compared the cumulative incidence of CTRCD for the two groups, using Kaplan-Meier curves, log-rank test, and univariate Cox proportional hazard model. To adjust for known risk factors and clinically relevant covariates, a multivariate Cox proportional hazard model was constructed to adjust for multiple clinical variables, including age as a continuous variable, sex, race (White, Black, Asian, or other), LVEF at baseline (as a continuous variable), history of coronary artery disease, hypertension, dyslipidemia, diabetes, obesity, initial anthracycline dose (as a continuous variable), and the presence of any ECG abnormalities. Of note, since only five patients whose self-reported ethnicity was Hispanic/Latino were included, an analysis regarding ethnicity could not be performed. ECG abnormalities included $1^{st}$-degree atrioventricular block, atrial fibrillation, bundle branch block patterns, ST-T change, premature atrial/ventricular contraction, low voltage in limb leads, and paced rhythm. Subgroup and sensitivity analyses were performed to evaluate the robustness of the model under various conditions. The subgroups tested include hematologic malignancies, solid tumors, male/female, baseline LVEF of ≤60%, >60%, anthracyclines of >40 mg/m², and ≤40 mg/m². Furthermore, sensitivity analyses regarding baseline LVEF (patients with baseline LVEF ≥ 55%, ≥50%, and ≥45%, instead of 60% in the main analysis), and regarding the cutoff of LVEF values to define the CTRCD diagnosis (cutoff LVEF of 55%, 50%, and 45%, instead of 53% in the main analysis) were performed. We also evaluated the model separately in data from BWH and MGH. Additional analyses excluding CTRCD within 30 days from the initial chemotherapy and limiting cases with at least one follow-up echocardiogram were conducted.

### Prediction model
We developed prediction models using the multivariable Cox proportional hazard model with or without AI-CTRCD score to show the independent contribution of the AI-CTRCD model over known risk factors. Variables included in the models were selected according to the clinical characteristics available at initiation of chemotherapy. Time-dependent AUROCs of the two models for 0.5, 1, 1.5, and 2 years after initiation of chemotherapy were calculated and then compared to investigate the additional discriminative value of the AI-CTRCD score in risk prediction of CTRCD[44,45]. Sensitivity and specificity as well as PPV and NPV of the model with the AI-CTRCD scores at two-year follow-up are reported. These calculations were conducted using *timeROC*[46] package (0.4) in R. The models' clinical utility was also examined in a deployment simulation cohort of 1000 patients treated with anthracyclines.

### Statistical analysis
The AI models were developed using Tensorflow 2.6[47] in Python 3.7.3. Kaplan-Meier curves were plotted using *survival* package (3.5.7) and *ggplot2*[48] package (3.4.4) in R 3.6.1. Cox proportional hazard analyses were also performed using *survival* package in R. Fine and Gray model was constructed using *cmprsk* package (2.2.11). Continuous variables were presented as mean ± SD, and categorical variables were described as numbers and percentages if not otherwise specified. A two-sided $p < 0.05$ was considered significant for all analyses.

### Inclusion and Ethics
This study complies with all ethical regulations and guidelines. This study protocol was approved by the institutional review board of Mass General Brigham (2019P002651) and Keio University School of Medicine (approval number: 20200030). The IRB of participating institutions provides a waiver for individual consent if the research meets the following criteria: the study involves no more than minimal risk to the subjects; the waiver or alteration will not adversely affect the rights and welfare of the subjects; the research could not practicably be carried out without the waiver or alteration; and whenever

appropriate, the subjects will be provided with additional pertinent information after participation. The IRB approved the waiver of consent since the study met these criteria. This study had minimal patient risk: it collected data retrospectively, there was no direct contact with patients, and data were collected after medical care was completed. The only minimal risk was the breach of confidentiality during data abstraction from the electronic health record system. It was considered impractical to perform the study with consent given the large number of participants and the need for using historical data where the patients were no longer in the system. Waiver of consent was considered essential to recruit an unbiased and representative cohort of patients.

### Reporting summary
Further information on research design is available in the Nature Portfolio Reporting Summary linked to this article.

## Data availability
The data that support the findings of this study are available in the article, in the Supplementary Information, and on request from the corresponding author R.C.D. upon approval of the data sharing committees of the respective institutions. The data are not publicly available due to the presence of information that could compromise research participant privacy. Data use agreement will be required for data sharing. Requests will be responded within 3 months by the corresponding author. Source data are provided with this paper.

## Code availability
The code for training and testing the model is provided at https://github.com/obi-ml-public/ECG-LV-Dysfunction. The model weights may contain personal information from patients and thus, are not shared. We provide a web interface to run our model and generate predictions at http://onebraveideaml.org.

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

## Acknowledgements

This work was supported by One Brave Idea, co-funded by the American Heart Association and Verily with significant support from AstraZeneca and pillar support from Quest Diagnostics. S.G. was supported by grants from SECOM Science and Technology Foundation, Gout and Uric Acid Foundation, Sakakibara Heart Foundation, the Drs. Morton and Toby Mower Science Innovation Fund Fellowship. R.Y. is partially supported by the Uehara Memorial Foundation. Y.K. was supported by JST Grant Number JPMJPF2101, and a grant from JSR corporation and Taiju Life Social Welfare Foundation, Kondou Kinen Medical Foundation and Research fund of Mitsukoshi health and welfare foundation.

## Author contributions

R.Y collected the data from MGB, designed the study, carried out model development and statistical analysis, and drafted the manuscript. S.G designed the study, carried out the statistical analysis and drafted the manuscript together with R.Y. Y.H, M.H and Y.K collected and analyzed the data from Keio. C.A.M made critical revisions to the manuscript. R.C.D collected data and co-drafted the manuscript.

## Competing interests

R.C.D. has received consulting fees from Novartis and Pfizer, and is co-founder of Atman Health. C.A.M. has received consulting fees from Foresite Labs, Clarify Health, Bayer, BioSymetrics, Dinaqor, Janssen, Affinia, Dewpoint Therapeutics, and Pfizer and is a co-founder of Atman Health. The other authors declare no competing interests.
