## [Peer Review File · Nature Communications]

Artificial Intelligence-Enabled Prediction of Chemotherapy-Induced Cardiotoxicity from Baseline ElectrocardiogramsREVIEWER COMMENTS

Reviewer #1 (Remarks to the Author):

Summary: The authors sought to evaluate a previously developed AI-EF model for detection of left ventricular dysfunction (LVEF <40%) based on 12-lead ECG data for prediction of rather than detection of chemotherapy related cardiac dysfunction (CTRCD) in a cohort of patients undergoing chemotherapy treatment with anthracyclines. They found the AI-EF model in conjunction with clinical variables to predict CTRCD with an AUC of 0.77 vs. 0.75 (for clinical variables alone). They also show a high AI-EF score to be associated with a higher risk of CTRCD following multivariable adjustments.

General comments: I would like to commend the efforts of the authors in attempting to address this important topic and their evaluation of AI-based ECG technologies which not only has important clinical implications but also the potential to improve cardiovascular care provided to patients undergoing chemotherapy. The article reads very well, and study rationale is well explained. However, I have a few concerns/questions detailed below.

Specific comments:

- It appears the goal was to predict underlying left ventricular pathology (subclinical or concealed) prior to initiation of chemotherapy. Given CTRCD is believed to be caused by the chemotherapeutic regimen, it seems unlikely that we will be able to predict this future event before the inciting factor is actually introduced. It brings up the question of - what is the model actually predicting? particularly if that patient does not go on to receive chemotherapy. I suspect the AI-EF model likely just predicts an overall poor LV substrate and susceptibility to any adverse exposures and as such is not necessarily specific for CTRCD and the conclusions by the authors might need to clarify/highlight this.
- Did the authors consider just using the model to detect the presence of CTRCD once it develops? Given they provide data that about half of patients undergoing chemotherapy do not get echocardiograms due to limited resources. ECGs could be easily obtained instead and if a positive AI screen is detected, then potential therapies can be initiated once a confirmatory echo is performed rather than the intervention being continued echo screening (probably over 2 years) given this is already what the guidelines recommend for

all patients undergoing chemotherapy with cardiotoxic drugs.

- For the specific prediction of CRTCD, perhaps the authors might need to consider training and evaluating a new model among patients who have actually begun chemotherapy. I realize the authors have pointed out a limitation related to sample size with doing this, but perhaps their ability to successfully collaborate with multiple institutions based on their publication regarding the AI-EF model derivation (ref 26) and potential use of generative adversarial networks could help address the limited sample size.
- The AUC of 0.77 vs. 0.75 seems to be a marginal improvement in performance with addition of the AI-EF to model to known risk factors. Although this difference was shown to be statistically significant, one can argue that it might not necessarily be clinically important. The clinical risk factors model appears to do well on its own (AUC 0.75). Can the authors provide an AUC for the AI-EF model by itself to see if using the ECG alone would be sufficient for prediction if other clinical variables/risk factors are not readily available.
- In the results section, the authors mention that 1,138 individuals were enrolled. Were these patients prospectively enrolled or did the authors simply mean that data on these patients were extracted from electronic health records? If the latter, would suggest re-wording to avoid confusion.
- Minor - Also in the results section, the original AI-EF model was reported to have an AUC of 0.92 but in the published paper and later in this manuscript, this appears to be 0.91 correct?
- The authors state that the "AI-EF model demonstrated excellent stratification of the risk of CRTCD" and I was wondering how this was determined? Based on the hazard ratio? an AUC of 0.77, a positive predictive value of 16.9%? I would not consider any of these 'excellent stratification' and this sentence seems overstated.
- For the sentence "adjustment for known risk factors including age, sex, cancer types, low baseline observed LVEF on echocardiogram, co-morbidities" in the abstract and result sections, I noticed that race was not included. Was this intentional as subsequent sentences and Fig 2 suggest that the authors also controlled for race.
- Also, is there a reason ethnicity was not described in addition to race?
- Did the authors consider sex-based stratification when evaluating the performance of the 2 models described given, we know that there are sex-based differences in cardiovascular outcomes?
- When evaluating the performance of the AI-EF model, did the authors make sure that the

patients included in the cohort presented in this manuscript were not included in the original training and validation cohorts given their prior paper (ref 26) discussing model derivation mentions that data was obtained from BWH, MGH, UCSF, and Keio.

- The description of the results stating that “the model including the AI-EF score consistently outperformed a model without the AI-EF score” and “the addition of the AI-EF score yielded a significant increment in predictive performance beyond conventional risk factors” both seem overstated as the differences shown in table 2 seem marginal. While the difference was statistically significant, it does not appear to be clinically significant. Would suggest rewording both sentences. It would be great if the authors provided other measures of diagnostic accuracy for side-by-side comparison of both models i.e. sens, spec, ppv, npv.
- The discussion suggests the performance of model 1 (including the AI-EF model) is similar to GLS alone (AUV 0.77) based on the literature. Did the authors consider potential subgroup analysis among those with GLS values to see if the AI model has additional value when compared to GLS or in combination with it?
- I see the patients in the study cohort were identified within 90 days prior to initiation of chemotherapy but what was the duration between the ECGs and echocardiograms in this analysis? As the further away the ECG is from the echocardiogram likely impacts its predictive value.
- The authors mention the use of a cutoff value determined by the Youden index. Was this the same cutoff value used in the derivation model (ref 26)? If yes please clarify by stating in the methods and if not, why was a new threshold used in this study and what was the rationale?

Reviewer #2 (Remarks to the Author):

This document evaluated the performance of an AI-ECG algorithm developed to detect an EF <40% for the prediction of anthracycline-related cardiomyopathy (decline in LVEF >10% to <53%). Starting out with a total of 5,476 patients received chemotherapy with a regimen including anthracyclines at Brigham and Women’s Hospital (BWH) and Massachusetts General Hospital (MGH) between June 1st, 2015, and October 1st, 2020, only 1,138 individuals had an ECG and echocardiogram within 90 days prior to chemotherapy, which constituted the study population. Over a median follow-up duration of 560 days, 99

participants (8.9%) experienced anthracycline-related cardiomyopathy. Compared to patients with low AI-EF scores (i.e., predicted as normal LVEF), patients with high AI-EF scores (i.e., predicted as low LVEF) were at higher risk of anthracycline-related cardiomyopathy (hazard ratio (HR), 2.14; 95% confidence interval (CI) 1.43-3.19; log-rank $p < 0.001$). This finding remained consistent after adjusting for multiple risk factors, including age, sex, cancer types, observed baseline LVEF, and overt ECG abnormalities (adjusted HR, 2.19; 95%CI, 1.40-3.41; $p < 0.001$).

This is a well written paper with appeal; a few points, however, are to be made:

- 1) 7.1% in the low AI-EF score group developed anthracycline-related cardiomyopathy vs. 8.9% overall and 12.9% in the high AI-EF score group; alternatively expressed, 41 of the 99 patients who developed anthracycline-related cardiomyopathy had a low-risk score. While the hazard model appears promising, the prognostic performance, the differential ability to stratify into truly low risk and truly high risk by the AI-ECG algorithm for low EF does not seem to be as strong.
- 2) Time-dependent AUCs at 0.5, 1, 1.5, and 2 years follow-up yielded values in the upper 70s for models with AI-ECG input, 2.8 points higher than models without AI-ECG input. Highest specificity and PPV reported in the main results section are 70.8% and 20.8%, respectively, sensitivity and NPV at this AUC cut-point 69.9% and 95.5%. This is consistent with point on the discriminatory ability of the AI-ECG and its additive value.
- 3) Retrospective datasets always pose a challenge; how many of the patients did have a pre- and post-therapy echocardiogram or EF assessment that would allow for robust adjudication of primary event rate?
- 4) For the reporting of the anthracycline dose, cumulative doxorubicin equivalents is the most commonly reported and might be worth changing.
- 5) Subgroup and sensitivity are useful and informative.
- 6) Mortality is all-cause mortality, any option to sub-stratify CV and non-CV, or cancer and non-cancer?

Response to Reviewers

We appreciate the overall positive assessment of our work given by the reviewers and thank them for their valuable comments. Your feedback has brought to our attention several areas that require improvement.

We have worked on improving our model using a new dataset from an additional hospital in Japan, and we have made significant changes to our paper based on the feedback from reviewers accordingly. Collaboration with researchers in Japan allowed us to develop a new AI model specifically trained to detect chemotherapy-related cardiac dysfunction (CTRCD). We have made major revisions to our tables, figures, and the main text of our paper accordingly, which we believe improved the clarity and validity of our findings.

We have listed the issues raised by reviewers in bold and our responses in plain type unless otherwise specified. Modification in the text is shown in italic text within this letter.

To Reviewer #1:

Summary: The authors sought to evaluate a previously developed AI-EF model for detection of left ventricular dysfunction (LVEF <40%) based on 12-lead ECG data for prediction of rather than detection of chemotherapy related cardiac dysfunction (CTRCD) in a cohort of patients undergoing chemotherapy treatment with anthracyclines. They found the AI-EF model in conjunction with clinical variables to predict CTRCD with an AUC of 0.77 vs. 0.75 (for clinical variables alone). They also show a high AI-EF score to be associated with a higher risk of CTRCD following multivariable adjustments.

General comments: I would like to commend the efforts of the authors in attempting to address this important topic and their evaluation of AI-based ECG technologies which not only has important clinical implications but also the potential to improve cardiovascular care provided to patients undergoing chemotherapy. The article reads very well, and study rationale is well explained. However, I have a few concerns/questions detailed below.

Thank you very much for the thorough review and precise understanding of our manuscript. We have extensively revised the manuscript according to your specific comments, as detailed below.

1. Specific comments:

- It appears the goal was to predict underlying left ventricular pathology (subclinical or concealed) prior to initiation of chemotherapy. Given CTRCD is believed to be caused by

the chemotherapeutic regimen, it seems unlikely that we will be able to predict this future event before the inciting factor is actually introduced. It brings up the question of - what is the model actually predicting? particularly if that patient does not go on to receive chemotherapy. I suspect the AI-EF model likely just predicts an overall poor LV substrate and susceptibility to any adverse exposures and as such is not necessarily specific for CTRCD and the conclusions by the authors might need to clarify/highlight this.

As the reviewer pointed out, the AI model is applied before the initiation of chemotherapy. Our aim was to detect baseline cardiac abnormalities that, in the context of the stresses of chemotherapy (a second hit), lead to CTRCD. We agree that it is unclear if the finding is specific to CTRCD or simply reflects heightened susceptibility to multiple adverse exposures, as is the case with some genetic mutations. We clarified this point as follows in the Abstract and the Discussion section.

Abstract

“Recently, an artificial intelligence (AI) model was shown to detect reduced left ventricular ejection fraction (LVEF) from 12-lead electrocardiograms (ECG) (AI-EF model), suggesting that the model identifies ECG features reflective of left ventricular pathobiology. This led us to hypothesize that an AI model can be trained to extract the baseline cardiac abnormalities that determine the susceptibility to cardio-toxic agents causing CTRCD by leveraging the information from the AI-EF model. To test this hypothesis, we trained an AI model to predict CTRCD from baseline ECGs obtained prior to the initiation of chemotherapy by applying the transfer learning approach to the AI-EF model and assessed the association of the AI score (AI-CTRCD score) with CTRCD.”

Discussion

“Since the AI-CTRCD model is applied before the initiation of chemotherapy, the model could only detect abnormalities present at baseline. Thus, the prediction of CTRCD probably is driven by the detection of susceptibility to cardiac damage induced by chemotherapy. While the detected features may not be specific to the susceptibility to chemotherapy and could be a general marker of a poor left ventricular substrate, we believe that the extraction of predictive features from a low-cost modality (i.e., ECG) is valuable. For instance, although GLS is an echocardiographic value assessing left ventricular systolic function and not necessarily specific to the risk of CTRCD, its clinical utility as a risk factor of CTRCD is widely recognized in cardio-oncology. Our results demonstrate the possibility of expanding the applicability of predictive measures over GLS by utilizing a much more accessible modality. “

We have also modified the conclusions as follows:

“Our findings support the clinical utility of baseline ECG before the initiation of chemotherapy, together with the AI-CTRCD model in identifying patients treated with anthracyclines at elevated risk of CTRCD.

2. Did the authors consider just using the model to detect the presence of CTRCD once it develops? Given they provide data that about half of patients undergoing chemotherapy do not get echocardiograms due to limited resources. ECGs could be easily obtained instead and if a positive AI screen is detected, then potential therapies can be initiated once a confirmatory echo is performed rather than the intervention being continued echo screening (probably over 2 years) given this is already what the guidelines recommend for all patients undergoing chemotherapy with cardiotoxic drugs.

Thank you for the insightful comment. We agree that the AI-EF model could also be applied to the CTRCD detection purpose. To this end, we have additionally collected the data of patients with both surveillance ECG and sequential echocardiographic test (within 30 days between the two procedures) after the first chemotherapy, and test whether the AI-EF model detected decreased LVEF. Results show that the AUROC of the AI-EF score and CTRCD was 0.875. However, only 9 out of 1138 patients were included under the abovementioned criteria of having both procedures within 30 days. This low number is expected as current guidelines do not recommend routine ECG tests after the initiation of chemotherapy. Thus while the reviewer proposes a potential workflow that would likely be successful with our tool, we anticipate there will rarely be the input data required. Therefore, we did not include these results in the manuscript.

Another possibility is to use the ECGs taken after the initiation of chemotherapy to predict CTRCD. ECGs taken after the initiation of chemotherapy could contain information about the damage caused specifically by the chemotherapy and thus could improve the prediction. To test this, we additionally collected ECGs obtained within a year after the first chemotherapy. We found that 787 (69%) of the overall population in the MGB cohort underwent ECGs within a year after chemotherapy initiation. Among them, 78 (9.9%) experienced CTRCD. Results showed that the AI-EF model similarly stratified the risk of developing CTRCD after the ECGs (HR 2.64). The result is shown in the reviewer-only material as follows:

However, the number of patients who underwent ECG after the initiation of chemotherapy was approximately 2/3 of the study population (787 vs. 1138, respectively). Furthermore, the prevalence of CTRCD in patients who had an ECG after chemotherapy was higher (9.9%) than that in patients who did not have an ECG after chemotherapy (6.4%). This indicates that a selection bias was inevitably introduced; since symptomatic patients, whom physicians likely consider to be at high risk for heart failure, undergo ECG tests more frequently compared to asymptomatic patients, whom physicians consider to be at low risk for heart failure. Given that follow-up ECG is not routinely performed or recommended by guidelines, it could be infeasible to determine whether the strategy validly works using a retrospective cohort from EHRs. We have clarified this issue in the Discussion section as follows:

“Another but more straightforward approach for early detection of CTRCD using AI-EF model is to utilize surveillance ECGs taken after the initiation of chemotherapy to detect patients who already develop CTRCD as early as possible. Since routine ECG after chemotherapy is currently not recommended, patients who underwent ECGs after chemotherapy likely had cardiac dysfunction because physicians considered them to be at high risk for heart failure (i.e., they presented subjective symptom), leading to a significant bias when analyzed retrospectively. Further prospective studies are warranted to validate this strategy to apply ECG for early diagnosis of CTRCD while it is asymptomatic.”

3. For the specific prediction of CTRCD, perhaps the authors might need to consider training and evaluating a new model among patients who have actually begun chemotherapy. I realize the authors have pointed out a limitation related to sample size with doing this, but perhaps their ability to successfully collaborate with multiple

institutions based on their publication regarding the AI-EF model derivation (ref 26) and potential use of generative adversarial networks could help address the limited sample size.

Thank you for providing us with your valuable advice. We agree that directly training an AI model using data from patients who receive cardiotoxic chemotherapy will improve the focus of the model and could potentially improve predictive performance. To evaluate this possibility, we additionally collected data on patients treated with anthracyclines from a Japanese hospital (Keio University Hospital). Since the number of recruited patients was still low, we could not develop a de-novo model for this task. However, after constructing a training cohort consisting of the Keio cohort and a part of BWH cohort, we re-trained our AI-EF model to predict CTRCD using baseline ECGs by transfer learning approach (AI-CTRCD model). The model was trained to predict CTRCD from baseline ECG specifically. Results show that transfer learning specifically targeting CTRCD improved the prediction. We extensively revised the manuscript, figures, and tables accordingly as below:

Abstract

“Recently, an artificial intelligence (AI) model was shown to detect reduced left ventricular ejection fraction (LVEF) from 12-lead electrocardiograms (ECG) (AI-EF model), suggesting that the model identifies ECG features reflective of left ventricular pathobiology. This led us to hypothesize that an AI model can be trained to extract the baseline cardiac abnormalities that determine the susceptibility to cardio-toxic agents causing CTRCD by leveraging the information from the AI-EF model. To test this hypothesis, we trained an AI model to predict CTRCD from baseline ECGs obtained prior to the initiation of chemotherapy by applying the transfer learning approach to the AI-EF model and assessed the association of the AI score (AI-CTRCD score) with CTRCD.”

Results

*“Similarly, a total of 880 cancer patients were treated with anthracyclines at Keio University Hospital between January 2013 and December 2019. Of those, 190 patients who underwent baseline ECG and TTE were included in the study (**Supplementary Table 3**). To create a training dataset, the BWH cohort was randomly split in a 2:8 ratio and the former (n=127) was combined with the Keio cohort (**Supplementary Fig. 1, Supplementary Table 4**).”*

*“Overall, participants in the MGB test set were 57.1±16.4 years old, and 47.8% were male (**Table 1**). Most cancer diagnoses were hematologic malignancies (n=704, 69.7%). The mean LVEF at baseline was 65.1±6.5%. While mean age was not different*

*between patients with and without CTRCD, baseline LVEF was significantly lower at baseline in patients who went on to develop CTRCD than in patients without CTRCD (age, 57.5±16.8 years and 57.1±16.3 years, P=0.97; LVEF, 62.4±6.5% and 65.4±6.4%, P<0.001 for patients with and without CTRCD, respectively). The prevalence of comorbidities was similar between the two groups. Patients at BWH were older than those at MGH, and a lower baseline LVEF and higher prevalence of leukemia were observed at BWH compared with MGH (**Supplementary Tables 1 and 2**)."*

*"Patients in the high AI-CTRCD score group were at higher risk for developing CTRCD compared to those in the low AI-CTRCD score group (incidence rate per 100 person-year, 8.93 vs 3.08 in high and low AI-CTRCD score groups, respectively; hazard ratio (HR), 2.66; 95% confidence interval (CI), 1.73-4.10; log-rank p<0.001; **Fig.1A**)."*

Methods

"Similarly, patient demographics data and prescription information were collected from Keio University Hospital through manual chart review with the same inclusion/exclusion criteria. ECG data were recorded and stored in a system provided by Nihon Kohden."

*"To update the model specifically for the CTRCD prediction purpose, we employed a transfer learning approach³⁷. The model training was done based on the Keio dataset, but to enhance the model's generalizability, the BWH cohort was randomly split in a 2:8 ratio and the former was combined with the Keio cohort to construct a training set (**Supplementary Fig. 1**). The training set was further randomly split into two groups (derivation and validation sets) in a 7:3 ratio. The weights of the abovementioned AI-EF model were used as initial weights of a new AI model with the same architecture, and the model was re-trained on the new training dataset to predict the future occurrence of CTRCD (AI-CTRCD model) with a learning rate of 0.00001 using RMSprop optimizer. The training process was completed in 150 epochs, and a final model with the highest AUROC in the validation set was chosen and tested once using the rest of the MGB test set."*

Also, we thank the reviewer for the suggestion of using the GAN. While the GAN approach can generate new images by learning the features from training data, the information content is reported to be not increased from the total amount of information in the original dataset.^{1,2} The training of the GAN model itself also suffers from the small number of sample available. Thus, after careful consideration, we employed a transfer learning approach using the AI-EF model to develop a new AI model that specifically focuses on the CTRCD detection task.

4-1. The AUC of 0.77 vs. 0.75 seems to be a marginal improvement in performance with addition of the AI-EF to model to known risk factors. Although this difference was shown to be statistically significant, one can argue that it might not necessarily be clinically important. The clinical risk factors model appears to do well on its own (AUV 0.75).

We agree with this reviewer's point that statistical significance in the AUC does not necessarily mean clinical significance. We would like to highlight that there is no established prediction model to stratify the risk of CTRCD, and thus, the model including all known risk factors is also not clinically available. The purpose of the comparison was to evaluate the independent contribution of the AI-CTRCD model against known risk factors for predicting CTRCD and was not to compare the clinical utility of these 2 models. We agree that this point was not clear in the previous manuscript. We have clarified the aim of this analysis in the method section as follows.

"We developed prediction models using the multivariable Cox proportional hazard model with or without AI-CTRCD score to show the independent contribution of the AI-CTRCD model over known risk factors."

In addition, we separated the section describing the model's clinical utility in the Results section as follows to clearly separate the analysis evaluating the incremental value of the AI-CTRCD score from the one evaluating clinical utility.

"The model, including the AI-CTRCD score, improves the detection of CTRCD under limited echocardiogram capacity.

*To test the clinical utility of the model, we performed a deployment simulation assuming a cohort of 1,000 patients receiving cardiotoxic chemotherapy with the same prevalence of CTRCD (9%) in a setting where all patients undergo baseline ECGs with the use of the full model (including the known risk factors and the AI-CTRCD model) to select patients on whom to perform follow-up echocardiograms. Based on the observed rate in our analysis, the resources for surveillance echocardiograms were assumed to allow 579 (57.9%) such procedures in the whole simulated cohort. In this context, 52 out of 90 patients with CTRCD would be detected if the follow-up echocardiograms were randomly provided. In contrast, if the target population was pre-selected based on our prediction model with the cutoff for a sensitivity of 93.5% and a PPV of 14.4%, 545 patients would be allocated to undergo surveillance echocardiograms (allowing for the echocardiographic evaluation of all patients tested positive from the ECG screening tool) and 84 out of 90 CTRCD patients would be detected (**Supplementary Table 6**). This*

simulation suggests that a predictive tool using the baseline ECG would result in a 60% higher detection rate (an additional 32 cases) in the detection of CTRCD with the same total number of echocardiograms performed. Similarly, when using a model with AI-CTRCD score and readily available clinical variables (age, sex, race, and cancer type), 546 patients would be categorized in the high-risk group, and 81 CTRCD cases (additional 29 cases compared to the random-echo strategy) would be detected at the cutoff of a sensitivity 90.5% with a PPV 14.8%.”

We also agree that the wording of statistical vs. clinical significance was not strict. We have rephrased “significance” throughout the manuscript to carefully clarified the statistical vs. clinical significance when we mention them.

Nevertheless, to answer the reviewer’s point, we have performed additional deployment simulations and added the results in Supplementary Table 5, as displayed below.

Supplementary Table 5

	Description	AUROC	Sensitivity	PPV	N of pre-screened patients	N of patients detected	N of additional cases detected
Model 1	Full + AI	78.1 (72.2-84.0)	93.5	15.4	545	84	32
Model 2	4 var + AI	75.0 (69.2-80.8)	90.5	14.8	546	81	29
Model 3	AI only	67.2 (60.3-74.0)	71.9	12.5	519	65	13
Model 4	4 var w/o AI	67.2 (61.5-74.4)	80.9	13.5	541	73	21
Model 5	Full w/o AI	73.8 (67.6-80.1)	87.5	14.6	541	79	27
Random		50	52.0	9	572	90	

AI: AI-CTRCD score, Full: age, sex, race, cancer type, initial anthracycline dose, past medical histories, 4 var: age, sex, race, cancer type

When comparing Model 1 (model using all clinical variables + AI-CTRCD score) with Model 5 (model using all clinical variables without AI-CTRCD score), Model 1 detected 5 more CTRCD patients within the echocardiogram capacity. While the incremental value may not be excellent, we believe that the detection of 5 additional patients (approximately 19% increase in detection) is not negligible.

4-2. Can the authors provide an AUC for the AI-EF model by itself to see if using the ECG alone would be sufficient for prediction if other clinical variables/risk factors are not readily available.

We agree that all clinical variables may not be available in some clinical settings (e.g., some patients may not remember their past history). Considering this situation, we calculated the AUROC of the AI-CTRCD model alone and compared models containing 4 readily available variables (age, sex, race, and type of cancer) with and without the AI-CTRCD model. The time-dependent AUROC at 2 years of the AI-CTRCD model alone was 0.672, and the AUROC at 2 years for the model, including age, sex, and cancer type, was 0.679. While the performance of both models was not sufficient, the AUROC of the 4-variable model with AI-CTRCD score had improved performance (AUROC 0.750), which is comparable to the performance of the full model. Furthermore, the 4-variable model with AI-CTRCD score showed similar performance in detecting CTRCD cases in the deployment simulation cohort. The 4-variable model with AI-CTRCD detects additional 8 cases compared to the 4-variable model without AI-CTRCD score. These results suggest that adding the AI-CTRCD model provides good prediction even when only limited information is available. We have additionally reported these results in **Supplementary Table 5** and modified the Methods and Results sections as follows.

Results

“Similarly, when using a model with AI-CTRCD score and readily available clinical variables (age, sex, race, and cancer type), 546 patients would be categorized in the high-risk group who have follow-up echocardiograms, and 81 CTRCD cases (additional 29 cases compared to the random-echo strategy) would be detected at the cutoff of a sensitivity 90.5% and a PPV 14.8% with the same echocardiogram capacity (Supplementary Table 5).”

Methods

“Furthermore, a model including AI-CTRCD score as well as readily available variables (age, sex, race, and cancer type) was developed using Cox proportional hazard model, and their diagnostic performance was reported.”

5. In the results section, the authors mention that 1,138 individuals were enrolled. Were these patients prospectively enrolled or did the authors simply mean that data on these patients were extracted from electronic health records? If the latter, would suggest re-wording to avoid confusion.

Thank you for pointing this out. The data were retrospectively extracted from electronic health records. We have reworded “enrolled” as follows in the Results section.

“Of these, 1,138 individuals who underwent ECG and transthoracic echocardiogram (TTE) within 90 days prior to the initial treatment with chemotherapy were retrospectively identified (Supplementary Fig. 1, Supplementary Table 1,2).”

We have further clarified this point in the “Methods” section as follows.

“Patient demographics data and prescription information were retrospectively obtained from the Massachusetts General Brigham (MGB) Enterprise Data Warehouse (EDW)”

6. Minor - Also in the results section, the original AI-EF model was reported to have an AUC of 0.92 but in the published paper and later in this manuscript, this appears to be 0.91 correct?

Thank you for pointing out our mistake. The correct value is 0.91 as the reviewer indicated and we corrected it on the manuscript.

7. The authors state that the “AI-EF model demonstrated excellent stratification of the risk of CRTCD” and I was wondering how this was determined? Based on the hazard ratio? an AUC of 0.77, a positive predictive value of 16.9%? I would not consider any of these ‘excellent stratification’ and this sentence seems overstated.

Thank you for providing us with your valuable advice. This statement was based on the hazard ratio of 2.66. We agree that “excellent stratification” was overstated and rephrased the expression as below:

“The AI-CTRCD model stratified the baseline risk of CRTCD using ECG.”

8. For the sentence “adjustment for known risk factors including age, sex, cancer types, low baseline observed LVEF on echocardiogram, co-morbidities” in the abstract and result sections, I noticed that race was not included. Was this intentional as subsequent sentences and Fig 2 suggest that the authors also controlled for race.

Thank you for pointing out our omission. Since we did include race in the adjustment but unintentionally did not mention it, we corrected the expression as below.

*“This finding was consistent after adjustment for known risk factors including age, sex, race, cancer types, baseline LVEF by echocardiogram, comorbidities such as coronary artery disease and hypertension^{2,25,27,28}, and the presence of overt ECG abnormalities (adjusted HR, 1.99; 95%CI, 1.26-3.16; p=0.003; **Fig 2**).”*

9. Also, is there a reason ethnicity was not described in addition to race?

Our dataset included information on patients' self-reported ethnicity. However, in the MGB test dataset, there were only 5 patients whose self-reported race/ethnicity was Hispanic/Latino. Thus we could not analyze the influence of ethnicity in our analysis. We additionally mentioned it in the “Methods” section as below:

“Since only 5 patients whose self-reported ethnicity was Hispanic/Latino were included, an analysis regarding ethnicity could not be performed.”

10. Did the authors consider sex-based stratification when evaluating the performance of the 2 models described given, we know that there are sex-based differences in cardiovascular outcomes?

Thank you so much for your thoughtful suggestion. We acknowledge the importance of sex on performance and have added a subgroup analysis stratified by sex. The model performed well regardless of the patients' sex. We have added the result in the “Result” section and main Fig. 3C, D as below.

*“Sex difference in cardiovascular outcomes is well recognized. However, a subgroup analysis based on patients' sex showed the consistent performance of the AI-CTRCD model (HR 2.54; 95%CI 1.41-4.58; log-rank p=0.001 and HR 2.70; 95%CI 1.43-5.12; log-rank p=0.002, respectively; **Fig 3C, D**).”*

11. When evaluating the performance of the AI-EF model, did the authors make sure that the patients included in the cohort presented in this manuscript were not included in the original training and validation cohorts given their prior paper (ref 26) discussing model derivation mentions that data was obtained from BWH, MGH, UCSF, and Keio.

Thank you for bringing this important problem to our attention. We have found that a very minor number of ECGs in the study population were shared with the original AI-EF model derivation cohort. Therefore, we developed a new AI-EF model using the same cohort without ECGs included in the study population, with equivalent AUROC (AUC 0.91 in the BWH test cohort), and used the new model for the transfer learning approach (described in the response to comment #3 for the same reviewer). We have clarified this point in the “Methods” section as below.

“Since a minor number of ECGs in the study population were included in the dataset for developing the AI-EF model, we developed a new AI-EF model after excluding these ECGs from the training dataset, and found the model had equivalent accuracy in detecting low LVEF from ECG (Supplementary Table 6).”

Supplementary Table 6. Performance of AI-EF model detecting low left ventricular ejection fraction from electrocardiogram used in the study

	BWH	MGH	UCSF	Keio
AUROC	0.91	0.89	0.91	0.92
(95%CI)	(0.89-0.93)	(0.86-0.91)	(0.88-0.93)	(0.90-0.93)

An AI model was trained after eliminating ECGs that were included in the CTRCD study population in the BWH cohort, and tested in the exactly same population (BWH, MGH, UCSF, Keio) in the previous study. BWH: Brigham and Women’s Hospital, MGH: Massachusetts General Hospital, UCSF: University of California San Francisco, Keio: Keio University Hospital)

12-1. The description of the results stating that “the model including the AI-EF score consistently outperformed a model without the AI-EF score” and “the addition of the AI-EF score yielded a significant increment in predictive performance beyond conventional risk factors” both seem overstated as the differences shown in table 2 seem marginal. While the difference was statistically significant, it does not appear to be clinically significant. Would suggest rewording both sentences.

We appreciate your valuable comment. We agree that the difference between “statistical significance” and “clinical significance” was not clear from our wording. Per the reviewer’s suggestion, we have rephrased these sentences below:

“The model including the AI-CTRCD score consistently showed statistically higher AUROCs compared with the model without AI-CTRCD score.”

“The addition of the AI-CTRCD score on conventional risk factors yielded a statistically significant increment in predictive performance over conventional risk factors”

12-2. It would be great if the authors provided other measures of diagnostic accuracy for side-by-side comparison of both models i.e. sens, spec, ppv, npv.

We have additionally calculated the diagnostic values of both the models with and without AI-CTRCD scores. We found that the model with the AI-CTRCD score had higher diagnostic accuracy compared to the model without the AI-CTRCD score at similar sensitivity levels. We have reported these results in the Results section, Table 3 as follows:

“The model with AI-CTRCD score detected CTRCD at 2 years with a positive predictive value (PPV) of 16.1% for a sensitivity of 91.1%, and with a PPV of 25.8% for a sensitivity of 60.5%, showing improvement from baseline prevalence (8.7%) (Table 3). These diagnostic values were improved compared to those from the model without AI-CTRCD score at the same sensitivity levels.”

Target sensitivity	Sensitivity		Specificity		PPV		NPV	
	Model 1	Model 2	Model 1	Model 2	Model 1	Model 2	Model 1	Model 2
100			0		8.7		NA	
90	91.1	92.3	47.8	30.3	16.1	12.7	98.0	97.3
80	80.6	82.4	60.4	50.1	18.2	15.3	96.6	96.3
60	60.5	60.6	81.2	70.7	26.1	18.5	94.9	94.3

13. The discussion suggests the performance of model 1 (including the AI-EF model) is similar to GLS alone (AUV 0.77) based on the literature. Did the authors consider potential subgroup analysis among those with GLS values to see if the AI model has additional value when compared to GLS or in combination with it?

Thank you for your valuable suggestion. We additionally reviewed the chart of the patients included in the study but found that GLS was not routinely measured or reported in our institutions. Thus, we could not perform additional subgroup analysis by GLS. Although we were unable to show that the AI model augmented CTRCD prediction beyond GLS values, this lack of GLS measurements could support the clinical utility of our study because GLS is not widely available in clinical practice. Nonetheless, since the point the reviewer raised is quite important, we have added this point to the limitation as follows:

“Since the GLS values were not available in our institutions, it remains unclear whether the AI-CTRCD model performs better compared to the prognostic value of GLS. However, the absence of GLS measurement in the real-world setting could emphasize the clinical utility of the AI-CTRCD model as a more accessible tool for baseline risk stratification of cardiotoxicity compared with GLS.”

14. I see the patients in the study cohort were identified within 90 days prior to initiation of chemotherapy but what was the duration between the ECGs and echocardiograms in this analysis? As the further away the ECG is from the echocardiogram likely impacts its predictive value.

Thank you for your insightful question. Per the reviewer’s advice, we have additionally assessed the impact of the days between baseline ECG and echocardiogram. We found that the median time difference is 3 days [IQR, 1-15 days], and the model performance was consistent when stratified by a subgroup by the time difference (<3 days and ≥3 days). We have additionally described this result in the “Results” section and in Supplementary Figure as below.

*“The median time difference between ECG and echocardiography procedures was 3 [IQR 1 -15.8] days. The model prediction was not influenced by the difference of the date (**Supplementary Fig. 3E, F**)”*

E Duration between ECG and TTE <3 days

F Duration between ECG and TTE ≥3 days

15. The authors mention the use of a cutoff value determined by the Youden index. Was this the same cutoff value used in the derivation model (ref 26)? If yes please clarify by stating in the methods and if not, why was a new threshold used in this study and what was the rationale?

Thank you for bringing this issue to our attention. Since we developed a new model (AI-CTRCD model), the comparison of the cutoff values of the AI-EF model between the studies was no longer relevant.

To reviewer #2:

This document evaluated the performance of an AI-ECG algorithm developed to detect an EF <40% for the prediction of anthracycline-related cardiomyopathy (decline in LVEF >10% to <53%). Starting out with a total of 5,476 patients received chemotherapy with a regimen including anthracyclines at Brigham and Women's Hospital (BWH) and Massachusetts General Hospital (MGH) between June 1st, 2015, and October 1st, 2020, only 1,138 individuals had an ECG and echocardiogram within 90 days prior to chemotherapy, which constituted the study population. Over a median follow-up duration of 560 days, 99 participants (8.9%) experienced anthracycline-related cardiomyopathy. Compared to patients with low AI-EF scores (i.e., predicted as normal LVEF), patients with high AI-EF scores (i.e., predicted as low LVEF) were at higher risk of anthracycline-related cardiomyopathy (hazard ratio (HR), 2.14; 95% confidence interval (CI) 1.43-3.19; log-rank p<0.001). This finding remained consistent after adjusting for multiple risk factors, including age, sex, cancer types, observed baseline LVEF, and overt ECG abnormalities (adjusted HR, 2.19; 95%CI, 1.40-3.41; p<0.001).

Thank you very much for the precise understanding and the favorable comment on our manuscript.

1. This is a well written paper with appeal; a few points, however, are to be made:

1) 7.1% in the low AI-EF score group developed anthracycline-related cardiomyopathy vs. 8.9% overall and 12.9% in the high AI-EF score group; alternatively expressed, 41 of the 99 patients who developed anthracycline-related cardiomyopathy had a low-risk score. While the hazard model appears promising, the prognostic performance, the differential ability to stratify into truly low risk and truly high risk by the AI-ECG algorithm for low EF does not seem to be as strong.

We appreciate your input and thank you for your time and effort in reviewing our manuscript. Since we developed a new AI model to specifically predict the risk of CTRCD (AI-CTRCD model), we re-calculated the incidence rate of CTRCD in low- and high-AI score groups. Results are reported as shown below.

“Patients in the high AI-CTRCD score group were at higher risk for developing CTRCD compared to those in the low AI-CTRCD score group (incidence rate per 100 person-year, 8.94 vs 3.09 in high and low AI-CTRCD score groups, respectively)”

As the reviewer indicated, the differential ability of the AI model alone could be insufficient. However, we suppose that the AI model would be used together with other patient information in clinical practice. To assess the differential ability of the AI model with clinical information, we additionally performed an analysis using the model, including the AI score and other variables (age, sex, race, cancer type, baseline LVEF, initial anthracycline dose, and past medical histories), showing better prediction of CTRCD. We found that the patients in the high-score group experienced CTRCD at a rate of 9.58 per person-year, whereas those in the low-score group had an incidence rate of 1.25 per person-year. Results are now shown in the Results section and Supplementary Fig. 5 as described below:

Results

*“Based on the full prediction model containing all the clinical variables and AI-CTRCD score, patients with high scores were approximately 7 times at higher risk of CTRCD compared to those with low scores (incidence rate per 100 person-year, 9.58 vs 1.25 in the high and low score groups, respectively; **Supplementary Fig. 5)**”*

Supplementary Fig. 5 Cumulative incidence of CTRCD stratified by the model with AI-CTRCD score and clinical variables

While the event rate in the low score group of 1.25 / 100 patient-years can be considered low, we agree that we could not argue that patients who were categorized as low-risk will not need echocardiographic follow-ups because missing patients with CTRCD could result in delayed initiation of cardioprotective therapy, that would lead to poorer outcomes. Instead, we propose that the AI model could effectively allocate echocardiograms within the current limitation of echocardiogram capacity by detecting those who benefit more from monitoring echocardiography after chemotherapy. Thus, we additionally clarified this point in the “Discussion” section as follows.

“Considering its impact on the poor outcome, it is crucial not to miss the cases with CTRCD. We proposed an AI-based approach using a single recording of baseline ECG as an instant tool to efficiently detect patients at high risk for CTRCD who should be monitored by echocardiography after chemotherapy within the current limitation of echocardiogram capacity.”

2. Time-dependent AUCs at 0.5, 1, 1.5, and 2 years follow-up yielded values in the upper 70s for models with AI-ECG input, 2.8 points higher than models without AI-ECG input. Highest specificity and PPV reported in the main results section are 70.8% and 20.8%, respectively, sensitivity and NPV at this AUC cut-point 69.9% and 95.5%. This is consistent with point on the discriminatory ability of the AI-ECG and its additive value.

Thank you for your generous remark. As the reviewer pointed out, we believe we were able to appropriately show that the gain in time-dependent ROC in the two years was consistently reflected in the improved diagnostic values of the model with the AI score.

3. Retrospective datasets always pose a challenge; how many of the patients did have a

pre- and post-therapy echocardiogram or EF assessment that would allow for robust adjudication of primary event rate?

Thank you for bringing this problem to our attention. We strongly agree that retrospective analysis could inevitably introduce bias. We found that 581 patients (57.5%) had both pre- and post-therapy LVEF assessment in the MGB test cohort. Among them, the strong association of the AI-CTRCD score and the risk of CTRCD was still observed (HR 2.43, 95%CI, 1.58-3.73, log-rank $p=0.0001$). Since this point the reviewer raised is quite critical to evaluate the validity of our analysis, we have added the description of this analysis in Results and clarified the limitation in the Discussion sections as follows:

“While 42.5% of the patients did not undergo a follow-up echocardiogram, a similar result was still observed after limiting the cohort to those with at least one follow-up echocardiogram (HR 2.43; 95%CI 1.58-3.73, log-rank $p<0.0001$) (Supplementary Fig. 3D).”

D Cases with follow-up echocardiograms

Discussion

“First, a number of patients were excluded because of a lack of ECG and/or echocardiogram prior to chemotherapy. Furthermore, CTRCD events could be missed with a lack of surveillance echocardiogram after chemotherapy. The study population might have been at a higher risk of CTRCD compared to the overall population receiving anthracyclines (physicians might have considered that these patients were likely to be at high risk for CTRCD and therefore performed ECG/echocardiographic evaluation). Nonetheless, this study indicated that the prediction model using the AI-CTRCD scores performed well in such a high-risk population.”

4. For the reporting of the anthracycline dose, cumulative doxorubicin equivalents is the most commonly reported and might be worth changing.

Thank you for providing a valuable suggestion. As the reviewer indicated, the cumulative doxorubicin equivalent dose is deeply associated with the occurrence of CTRCD, so we additionally reported it in Tables. Furthermore, an additional multiple Cox proportional hazard analysis using cumulative dose instead of initial dose shows an equivalent result (adjusted HR of AI-CTRCD score, 1.97 [95%CI 1.24-3.12]). At the beginning of the chemotherapy, however, it might be difficult to correctly estimate the cumulative dose that is finalized after the series of chemotherapy, unlike the initial dose. Therefore, we use the initial dose of anthracyclines for our prediction model. We have clarified this point in the “Results” section as follows:

“A high cumulative anthracycline dose is associated with the occurrence of CTRCD; however, the cumulative dose cannot be determined at the initiation of the chemotherapy and is thus not relevant for the prediction of adverse outcomes. Therefore, we performed stratification based on the initial anthracycline dose adjusted for body surface area as a surrogate.”

According to the reviewer’s comment, we acknowledge the importance of evaluating the impact of the cumulative dose of anthracyclines on the model performance. We found that the association of AI-CTRCD score and the risk of CTRCD was consistent in subgroups by cumulative dose (HR 2.82 (95%CI, 1.54-5.17) in the subpopulation of patients with cumulative anthracycline dose >180 mg/m² and 2.41 (95%CI, 1.30-4.49) in the subpopulation of patients with cumulative anthracycline dose ≤180 mg/m²). We have additionally described this result in the Results section as follows:

*“Results were also similar when the population was stratified by cumulative anthracycline dose (HR 2.82; 95%CI 1.54-5.17; log-rank p<0.001 and HR 2.41; 95%CI 1.30-4.49; log-rank p= 0.003 in patients receiving >180mg/m² and ≤180mg/m² of cumulative anthracycline dose, respectively; **Supplementary Fig. 3A, B)**”*

Supplementary Fig 3: Sensitivity analyses regarding patient characteristics.

Kaplan-Meier plot showing the cumulative incidence of CTRCD between high and low AI-CTRCD score in (A) patients who received >180mg/m² of cumulative anthracyclines (B) patients who received ≤180mg/m² of cumulative anthracyclines

5. Subgroup and sensitivity are useful and informative.

Thank you so much for your generous comment. We believe that the reported diagnostic values help readers understand how the model can contribute to the current cardio-oncology practice.

6. Mortality is all-cause mortality, any option to sub-stratify CV and non-CV, or cancer and non-cancer?

Thank you for your valuable suggestion. We additionally reviewed the chart of the patients included in the study, but found that there were 20 CVD death cases out of 375 patients who died during the study period. Since the number was small, we were not able to perform a valid stratifying analysis regarding the cause of death. Further studies with the larger number of patients and longer follow-up periods are warranted to address this issue. Nonetheless, the issue the reviewer raised is important from a clinical standpoint, we have noted this point in the limitation section below:

“Fifth, because only 20 CVD-related deaths in 375 patients who died during follow-up were seen in the MGB test set, the association of AI-CTRCD score and CVD-related death could not be assessed. Further studies with a larger population and longer follow-up periods are also warranted to show that the AI model could predict more severe cardiac dysfunction causing CVD death beyond the occurrence of CTRCD.”

References:

1. van den Oord, A., Li, Y. & Vinyals, O. Representation Learning with Contrastive Predictive Coding. *arXiv [cs.LG]* (2018).
2. Zhu, F., Ye, F., Fu, Y., Liu, Q. & Shen, B. Electrocardiogram generation with a bidirectional LSTM-CNN generative adversarial network. *Sci. Rep.* **9**, 6734 (2019).

REVIEWERS' COMMENTS

Reviewer #3 (Remarks to the Author):

The Authors are to be praised for the very actual topic of AI in Cardio-Oncology.

I don't have major concerns, I believe that the Authors have assessed the comments of Rev 2

Still, the Authors should update the reference list with more recent papers.

8. Instead of Ponikowski 2016 please update with the more recent ESC 2021 Guidelines: 2021 ESC Guidelines for the diagnosis and treatment of acute and chronic heart failure: Developed by the Task Force for the diagnosis and treatment of acute and chronic heart failure of the European Society of Cardiology (ESC). With the special contribution of the Heart Failure Association (HFA) of the ESC.

Authors/Task Force Members:; McDonagh TA, Metra M, Adamo M, Gardner RS, Baumbach A, Böhm M, Burri H, Butler J, Čelutkienė J, Chioncel O, Cleland JGF, Coats AJS, Crespo-Leiro MG, Farmakis D, Gilard M, Heymans S, Hoes AW, Jaarsma T, Jankowska EA, Lainscak M, Lam CSP, Lyon AR, McMurray JJV, Mebazaa A, Mindham R, Muneretto C, Francesco Piepoli M, Price S, Rosano GMC, Ruschitzka F, Kathrine Skibelund A; ESC Scientific Document Group. Eur J Heart Fail. 2022 Jan;24(1):4-131. doi: 10.1002/ejhf.2333.

PMID: 35083827

9. Instead of the very outdated Zamorano 2016 please reference the first ESC Cardio-Oncology guidelines 2022:

2022 ESC Guidelines on cardio-oncology developed in collaboration with the European Hematology Association (EHA), the European Society for Therapeutic Radiology and Oncology (ESTRO) and the International Cardio-Oncology Society (IC-OS).

Lyon AR, López-Fernández T, Couch LS, Asteggiano R, Aznar MC, Bergler-Klein J, Boriani G, Cardinale D, Cordoba R, Cosyns B, Cutter DJ, de Azambuja E, de Boer RA, Dent SF, Farmakis D, Gevaert SA, Gorog DA, Herrmann J, Lenihan D, Moslehi J, Moura B, Salinger SS, Stephens R, Suter TM, Szmit S, Tamargo J, Thavendiranathan P, Tocchetti CG, van der Meer P, van der Pal HJH; ESC Scientific Document Group.

Eur Heart J Cardiovasc Imaging. 2022 Sep 10;23(10):e333-e465. doi: 10.1093/ehjci/jeac106.
PMID: 36017575

25. Instead of the very outdated Plana 2014, please reference the more recent:

Role of cardiovascular imaging in cancer patients receiving cardiotoxic therapies: a position statement on behalf of the Heart Failure Association (HFA), the European Association of Cardiovascular Imaging (EACVI) and the Cardio-Oncology Council of the European Society of Cardiology (ESC).

Čelutkienė J, Pudil R, López-Fernández T, Grapsa J, Nihoyannopoulos P, Bergler-Klein J, Cohen-Solal A, Farmakis D, Tocchetti CG, von Haehling S, Barberis V, Flachskampf FA, Čėponienė I, Haegler-Laube E, Suter T, Lapinskas T, Prasad S, de Boer RA, Wechalekar K, Anker MS, Iakobishvili Z, Bucciarelli-Ducci C, Schulz-Menger J, Cosyns B, Gaemperli O, Belenkov Y, Hulot JS, Galderisi M, Lancellotti P, Bax J, Marwick TH, Chioncel O, Jaarsma T, Mullens W, Piepoli M, Thum T, Heymans S, Mueller C, Moura B, Ruschitzka F, Zamorano JL, Rosano G, Coats AJS, Asteggiano R, Seferovic P, Edvardsen T, Lyon AR.

Eur J Heart Fail. 2020 Sep;22(9):1504-1524. doi: 10.1002/ejhf.1957. Epub 2020 Aug 21.
PMID: 32621569

Response to Reviewers

We appreciate the positive assessment of our work given by the reviewer and thank them for their valuable comment.

To reviewer #1:

The Authors are to be praised for the very actual topic of AI in Cardio-Oncology. I don't have major concerns, I believe that the Authors have assessed the comments of Rev 2. Still, the Authors should update the reference list with more recent papers.

Thank you so much for the positive comments on our work and valuable suggestions regarding references. We have updated these references in the manuscript accordingly.